# Arithmetic vs. Weighted Means in Fish Fillets Mercury Analyses

**DOI:** 10.3390/ijerph21060758

**Published:** 2024-06-10

**Authors:** Helvi Heinonen-Tanski

**Affiliations:** Department of Environmental and Biological Sciences, University of Eastern Finland, P.O. Box 1627, FI-70211 Kuopio, Finland; helvi.heinonentanski@uef.fi

**Keywords:** food contamination, health protection, methyl mercury, sulfate reduction, weighted mean

## Abstract

Mercury (Hg) analyses in species of fish are performed for two reasons: (1) to safeguard human health; and (2) to assess environmental quality, since different environmental changes may increase the Hg concentrations in fish. These analyses are important since both natural and human activities can increase these Hg concentrations, which can vary extensively, depending on the species, age and catching location. Hg-contaminated fish or other marine foodstuffs can be only detected by chemical analysis. If the aim of Hg analysis is to protect the health of marine food consumers, researcher workers must consider the location where the fish were caught and interpret the results accordingly. Health and environmental officials must appreciate that in specific places, local people may have a daily diet consisting entirely of fish or other marine foods, and these individuals should not be exposed to high concentrations of Hg. Regional and national health and environmental officials should follow the recent guidance of international organizations when drawing their final conclusions about whether the products are safe or unsafe to eat. Correct statistical calculations are not always carried out; so, too high Hg amounts could be presented, and fish eaters could be protected. This work has been conducted to show the differences in Hg concentrations between weighted (weighted with fish weights) and arithmetic means. Thus, the mean that is only weighted also includes the Hg content in fishes; so, the exposure to Hg can be evaluated.

## 1. Introduction

Hg analyses in fish start from a selection of the fishing areas where there are suspicions about the Hg concentrations and possibly other contaminations in the marine environment, which will then be fished. Fresh or frozen fishes are then subjected to chemical analyses most often undertaken by highly trained personnel using expensive equipment [1], with the analyses typically being performed in a few accredited laboratories [2] at high costs. The last part involves the interpretation of the values with a valid statistical evaluation to allow for the drawing of reliable conclusions regarding whether fishes are safe or not safe.

Mercury toxicity became evident after some 80 years’ use of Hg compounds as medical compounds and as horticultural and agricultural pesticides [3,4]. The Minamata (Japan) tragedy was an alarm call since Hg compounds released from a battery-producing factory heavily contaminated the fish and other marine organisms in Minamata Bay in the south-western coast of Japan [5]. It caused serious nervous diseases in hundreds of people and even some deaths. Thus, in 1959, the Japanese Ministry of Health and many international and national organizations started efforts to ensure that this disaster could never again happen. The first restrictions and prohibitions were extended to different Hg compounds [6,7]. For example, the World Health Organization (WHO) and the Food and Agriculture Organization (FAO) set limits for how much Hg could be present in fish intended for human consumption [8]. It was appreciated that both marine and freshwater fishes and other products from the sea were the most critical Hg sources in the human diet. It was evident that the muscles of some large predatory fish tended to contain more Hg than those of fish which were herbivores. Two other factors found to be important were the industrial contamination of a fishing site and the size of the fish to be consumed [8]. 

Different organizations started to develop analytical methods to detect both inorganic Hg and methyl Hg. After the Minamata tragedy, the levels of Hg compounds in fish and other marine food have been monitored, especially near industrial enterprises, because too high mercury concentrations in different foods can be a serious obstacle for marketing these products. Therefore, commercial enterprises also became concerned about Hg and started to analyze its concentrations in their own specific products.

Today, the European Union (EU) has set two regulations, 2022/617 and 2023/915 [9,10], concerning the analysis of Hg in food products. The present regulations state that the fresh weight of fillets of most edible fish species can contain a maximum of 0.5 mg/kg total Hg, but for some very commonly used fish products such as sardines, anchovies and herrings, the limit is lower, only 0.3 mg/kg Hg. In contrast, some predatory fishes can contain a higher level of 1.0 mg/kg Hg. Today, in many countries, the recommendation is that humans should only seldom (or, in some cases, never) consume those fish species which can contain the highest concentrations of Hg.

### 1.1. Sulfate Reduction and Methyl Mercury Formation

The formation of methyl mercury has been known for a long time, e.g., in the 1990s, the first publications described that organic methyl mercury compounds could be formed during sulfate reduction [11,12]. Many anaerobic-sulfate-reducing bacteria possess an ability to produce methyl mercury as a side reaction from inorganic Hg compounds [12]. This reaction happens in freshwaters, marine waters, and wetlands both in tropical and arctic climates. The reaction can occur if inorganic Hg and sulfate are present at a low redox potential. The layers between water and sediments are thus good places for methyl Hg formation [12,13].

Sulfate-rich wastewaters accumulate in the bottoms of waterbodies because their specific weight is higher than that of surface waters with low sulfate concentrations. Thus, the bottom layers of waterbodies and, for example, under stones or in gravel and sand, will be the locations with the highest sulfate concentrations. An increasing concentration of sulfate decreases the redox potential and increases the risk of a greater reduction of inorganic Hg into methyl mercuric compounds [12,13]. It is evident that global mercury deposition is sufficient to allow for Hg methylation. It is known that the Hg can originate from volcanic eruptions or from human activities such as coal burning, gold mining using elemental Hg, etc. Similarly, the level of sulfate needed to allow for sulfate reduction and methyl mercury formation can be as low as 20–40 mg/L sulfate, and this level can originate from industrial emissions or from fertilizers [13,14]. Unfortunately, lake waters just above the sediments are seldom studied and, at least in Finland, they do not belong to obligatory analytical sites in environmental monitoring.

### 1.2. Toxicity of Methyl Hg

The formed methyl Hg compounds are more toxic to humans than the corresponding inorganic compounds [15] and the exposure dose of methyl Hg must always be considered. Methylated Hg compounds have a clear tendency to bioconcentrate via the following routes: bacteria, other microflora, small fishes, larger fish and mammals (including humans), and birds. Bioconcentration happens because the methyl Hg compounds are more fat-soluble than inorganic mercury compounds and, thus, they remain stored in fat tissues and are not available for metabolism and excretion. Therefore, the European Union Food Safety Authority [15] has issued (and updated) its scientific opinion about methyl mercury.

The three major messages of this scientific opinion report [15] are:As much as 90% of total Hg in fish can be present as methyl mercury.Small children and pregnant or lactating women and even women who can potentially become pregnant constitute the most sensitive individuals and they should avoid eating fish rich in methyl mercury.The safe weekly dose of methyl mercury for one person is proposed to be only 1.3 μg/body weight (kg).

The protection of pregnant women is important since Hg compounds are teratogenic. Furthermore, with respect to infants, the weekly Hg limitation means that a small child with a body weight of 10 kg could consume only 0.013 mg (13 μg) methyl Hg, i.e., a child should eat no more than 29 g of fish per week if one assumes that the fish would contain 0.5 mg/kg total Hg, of which 0.45 mg/kg would be methyl mercury. In practice, it means that families with small children and/or pregnant women must be confident that the fish on their dinner table should contain only low amounts of Hg and the fish should originate from an area known to be safe. Furthermore, the family should not eat fish too often even if the children themselves have participated in the catch.

The protection of fish consumers is possible if Hg and methyl Hg analyses are carried out and the amounts of Hg are known. The major aim of this paper is to show which type of mean counting method should be used to evaluate the true amount of Hg and health risk of Hg concentrations of perch intended for human consumption. It should be possible to analyze the total Hg amounts, which the fish eater can be exposed to, and methyl Hg can be estimated to be 90% of the total Hg [15]. The trustful results could also help fishers who sell a part of their catch to companies since these fish need to fulfill all legal requirements (including those related to Hg concentration), which may be analyzed in foreign customs laboratories. A major problem is using an appropriate way to determine if the edible fish contain levels of Hg too elevated so that fish-eaters can be protected.

## 2. Materials and Methods

### 2.1. Fish Analyses

The Water Framework Directive of European Union [16] states that all European waters should have at least a good ecological status and thus the fish and the other living organisms should be acceptable for human consumption. However, it is evident that the Hg concentration in fish depends on their living area, position in the food chain, size, and age [17,18]. Thus, the Hg concentration in fishes can vary extensively—but mainly it varies according to the size and age (weight and length) of the fish.

Analyses of mercury should be conducted in an accredited laboratory which should adhere to the protocols listed in the reference texts [9,10]. The analytical values presented here were performed by the Eurofins Environmental Testing Finland Oy, which is an accredited laboratory participating in quality control analyses with the same specimens being assayed in many laboratories in different countries. There are several stages to the evaluation of Hg concentrations in fish, e.g., the dissection of the sample to obtain fillet specimens, followed by Hg extraction, with the final analysis performed by gas chromatography coupled to mass spectrometry. There is always a degree of uncertainty associated with each Hg determination (as is the case in all chemical determinations) and the uncertainty found for some Hg assays can be as high as ±25%. The EU Commission’s regulation [19] offers some assistance about this problem although it should be included in the recommendations.

### 2.2. Chemical Analyses and Data Counting for Hg

The fishes were frozen and kept frozen until the Hg analyses so that volatile Hg compounds would be preserved. The determination of metals in fishes included Hg and some other metals (these concentrations were low, and these data are not presented in this work; furthermore, there are no limits set by European regulations) [15]. The analytical values presented here were performed by the Eurofins Environmental Testing Finland Oy, which is an accredited laboratory.

In mathematics, the mean of Hg concentration can be calculated in three ways; (1) as a simple arithmetic mean (average), (2) as a geometric mean, or (3) as a weighted mean (according to the fish’s weight). The arithmetic mean counts only the mean of Hg concentrations in different fishes. The geometric mean counts the mean after logarithmic formation. Arithmetic or geometric means do not consider the amount of Hg; thus, they do not warn of too high exposure Hg concentrations.

The weighted mean counts first the sum of Hg amounts in all fishes and this sum is then divided by the sum of the fish of all weights. The next equation shows the detailed way to count Hg amounts in fishes.
X = ∑w × c/∑w
where X is the weighted mean and ∑w × c is the sum of the weights for all fishes multiplied by the corresponding Hg concentrations and ∑w is the sum of weights of all fishes. Thus, ∑w × c also presents the total amount of Hg in all fishes.

The international regulations do not describe the exact detailed ways to calculate the final results [9,10]. These recommendations do not inform the health inspectors of what should be done if many but not all the analyzed fishes contain more than the limit value of Hg or if most of the edible fish fillets do contain more than the limit value but the arithmetic means are lower than the limit value. This phenomenon can happen, especially if there are large variations in the size of the fishes, i.e., there are many small-sized fishes which contain only low concentrations of Hg but there are some large ones which contain high concentrations of Hg.

Besides fish analyses, the same Eurofins laboratory also analyzed lake waters from the surface, middle and 1 m from the bottom using standard methods. These analyses included sulfate and other physicochemical parameters, but only sulfate concentrations are discussed here.

### 2.3. The Lakes

The selected lakes, Laakajärvi and Kiltua, are situated in the southern side of the Talvivaara mining area (today belonging to Terrafame company) in the eastern part of central Finland (app. 63°50′ N, 27°50′ E). Both these lakes are situated outside of the mining area; so, the fish should be assumed to be safe for human consumption, and the mining company has no permission to contaminate these lakes via its emissions. Lake Laakajärvi is the first waterbody downstream after the industrial mixing zone of the Terrafame mining area. Lake Kiltua lies downstream of Lake Laakajärvi; there are short rivers connecting Laakajärvi and the mining area’s water reservoir, and between Lake Laakajärvi and Lake Kiltua. In Lakes Laakajärvi and Kiltua, the levels of Hg and other values in fishes and water are set by the Water Framework Directive [16]. The area of Lake Laakajärvi covers 34.7 km^2^; Lake Kiltua is smaller, covering 10.1 km^2^.

### 2.4. The Used Report

Fishing for this work was carried out in autumn 2014 for the Talvivaara company (today, mining belongs to Terrafame) as a part of its obligatory environmental control. According to Finnish Environmental Law [20], these data must be made available to all people living in the mine’s vicinity, including fishers and fish consumers, who require this information about the quality of the fish catch and other environmental issues. The report used in this work originates from 2014 [21], since this is the last published report which also included data about all individual fishes, including their sizes and Hg concentrations. The report [21] was paid for, edited, and finally approved by Terrafame (or Talvivaara) before its release to the local environmental officials, who were then tasked with either accepting or rejecting the report’s findings. This report made it possible to calculate the arithmetic, the geometric and the weighted means (weighted according to each individual fish’s weight). The newer reports present only the arithmetic means.

The only fish species selected for this work is perch (*Perca fluviatilis*) since the number of northern pikes (*Esox lucius*) was often too low (Table 1). Nonetheless, perch is a suitable choice, since it is a popular fish with a good taste, and it can be easily fished even by children and families for their own consumption. One major problem with the very small perches with very small fillets is that it is laborious to obtain pure fillets after the removal of the fish’s scales, bones, and gut, without the spillage of bile.

The results of Table 2 were calculated with Microsoft Excel v2019.

## 3. Results

The results of the individual perch sizes and Hg concentrations are shown in Table 1. The original data are published only in Finnish [21]. The author has set the data in a new order so that the results may be easier to discern, and the four separate tables listed in [21] were merged into Table 1.

The sulfate concentrations in water in some previous years are presented in another report [22]. The concentration of sulfate was 628 mg/L in the bottom of Lake Laakajärvi in the autumn of 2014. The sulfate concentration in the bottom of Lake Kiltua was 37 mg/L in the autumn the 2014.

## 4. Discussion

### 4.1. The Different Means

The arithmetic means and geometric means for Hg are near 0.5 mg/kg. However, the arithmetic or geometric means do not give any realistic information about the actual amounts of Hg that the fish consumers were exposed to, if some fish contained only trace amounts of Hg, or that the large fishes had very high concentrations. It is impossible to estimate the exposure of Hg from either the arithmetic or geometric mean, which is crucial when one is estimating human exposure [15].

It should be noted, however, that the standard deviation, especially that in Lake Laakajärvi, was remarkably high, indicating that the fishes varied extensively in several parameters. This is evident from Table 1, where there were large variations in both the sizes and the Hg concentration in the perch, and all larger fishes contained more Hg than their smaller counterparts, a fact that has been previously reported internationally [15]. In addition, the correlation coefficients between the length of the fish and their Hg contents and between weight and Hg contents are high, again in accordance with previous publications [8,11,13,14].

The weighted means are the highest of all means (Table 2) since all the larger perches, from which good-sized fillets could be made, also contained the highest amounts of Hg. Thus, in Lake Laakajärvi, the smallest five perches (all with Hg concentrations of less than 0.5 mg/kg) represented less than 12% of total perch fillet mass, and, correspondingly, almost 89% of perch mass in the five larger fishes contained more than 0.5 mg/kg Hg (Table 2).

In Lake Kiltua, the three smallest perches accounted for some 21% of perch mass, while the percentage of the perch mass which contained over 0.5 mg/kg Hg was almost 79%; however, in two of these specimens, the level of Hg was only slightly above the proscribed limit and their values may not be considered due to methodological uncertainties. In fact, in both lakes, the fish in this “larger” category were not particularly big with respect to how large fish in this species can be; nonetheless, their Hg concentration did exceed the set limit, even considering methodological uncertainties.

It is often estimated that some 40% of fish weight is the edible part in fillets [23]. In Lake Laakajärvi, all perches considered for this work had a common weight of almost 2.2 kg (Table 2), and thus these fishes could be turned into about 880 g of perch fillets for food. This fillet mass could be enough for one substantial meal for five or six persons if a portion is estimated as 150 g, as presented in [23]. The consumers would thus be exposed to around 1620 μg of total Hg and around 1460 μg of methyl mercury, if 90% of Hg would be considered to be in methyl mercury form [15]. If this were to be their only weekly fish dinner, then the only way their weekly exposure dose of methyl mercury could be limited to 1.3 μg per kg of body weight [15] is if that the total weight of those six people eating the perch should be 1120 kg, which is most unlikely.

Both the total Hg and methyl mercury levels were lower (1010 μg and 909 μg) in Lake Kiltua perches since the total weight of these ten perches was 1.73 kg, giving around 700 g in fish fillet form. By using the same calculation, five people could consume these fillets. If that were to be their only fish dinner for a week, in order for them not to be exposed to an excess of Hg (and its methyl Hg form), their total body weight would need to be almost 700 kg.

In both these cases, the safe amount of perch fillet that should be served would be much smaller than what is commonly consumed by these lakeside inhabitants (150 g). In both these lakes, the mass of perch fillet on the plates should be reduced to less than 100 g per dinner.

If one critically examines the different means and their ability to protect the health of fish consumers against Hg risks, it is difficult to see any reason for utilizing the arithmetic mean, since the arithmetic mean does not consider the size of the fish and thus the Hg content of the portion of fish on the dinner plate. Only the weighted mean provides an indication of the total Hg exposure; this is clearly mentioned in an EU report [15] and in the Codex Alimentarius guidelines of FAO and WHO [24]. Both these international reports, made by groups of international experts, strongly emphasize the need to give details of both the total exposure and the maximum weekly intake of Hg (or other pollutants). In fact, the weighted mean has also been used in several scientific publications which have evaluated the exposure by diverse human groups to different pollutants [25,26,27,28]. The concept of the weighted mean is also applied when considering the different nutrients in human food or animal feed such as the consumption of proteins, carbohydrates, fatty acids, and different vitamins.

Thus, details of the weighted mean of Hg would protect the fish-eating population better than the arithmetic mean, or geometric mean despite the fact it gives the highest Hg concentrations. The weighted mean takes into account the contribution of larger fishes, as these will make up the greatest amounts of actual fish on the plate if the entire catch were to be consumed.

The result is not very trustful if the weight of perches (in this case) varies from 41 g to 540 g, meaning that the Hg concentrations also vary highly, and the numbers of pikes is so low that they should be omitted. The companies who must present fish results must take enough time to carry out careful fishing so that the numbers of fishes are at least ten (if that is the claim). In addition, the size of all fishes of the same species must be moderate and typical for those fishes which will be consumed.

### 4.2. Why the Fishes in These Lakes Contained Too Much Mercury

We can speculate about the reasons for the excessively high Hg levels in fish in these lakes. The general atmospheric fallout of Hg is also an evident Hg source in this part of Finland [11,12]. The sulfate concentrations were high at the bottoms of both lakes in the report of the third annual cycle [22]. A sulfate concentration of 628 mg/L is very high in comparison with the maximal sulfate concentration of 12 mg/L SO_4_ in 36 Finnish forest streams monitored for 20 years [29]. Evidently, the high sulfate concentration in the bottom of lake waters is a sign that there was an unsuccessful complete mixing of the different layers of the lake’s water layers, i.e., oxygen in the surface layers was not transported to the bottom (typically, these lakes are dimictic, with two annual complete mixings—in spring and in autumn due to temperature differences in the water layers). Due to disturbances in the full water mixings, there are high sulfate concentrations in the bottom of Lake Laakajärvi and the most sulfate-rich water with the highest specific weight survived in bottom of Lake Laakajärvi, allowing for the methylation of Hg and causing high Hg concentration in fishes.

In Lake Kiltua, the highest sulfate concentration in the bottom layer of water was 37 mg/L, which may indicate that complete mixings of the waters may have occurred in spring and/or autumn, and therefore the methylation of Hg was lower.

A massive accidental leakage from the Talvivaara mining area occurred in 2012 between 4 November and 15 November when at least 200,000 m^3^ of highly contaminated gypsum precipitation solution spread into the natural waters and forests towards Lakes Laakajärvi and Kiltua. During this event, high amounts of acids were formed, i.e., the pH measured in the leaking wastewater was at pH 2.89 instead of pH 9, and this polluted water contained high concentrations of heavy metals and sulfate [30]. It is likely that sulfate-rich wastewater continued to reach Lakes Laakajärvi and Kiltua.

### 4.3. The Local People and Animals as Fish Eaters

Many farmhouses, other permanent dwellings and summer cottages are situated along the shores of Lakes Laakajärvi and Kiltua. Some families have lived in this area for generations and, for these people, a fish diet was their staple. Fish still represents the major source of Hg throughout Finland [31]. In addition, while many local fishers may sell a part of their catch, others may eat the fresh fish from their home lake many times a week, especially if they catch a large fish. It is evident from the published report [21] that there were no truly large perch, although specimens as large as 1 kg can be caught, and as stated, it is the largest fish which contain the highest concentrations of Hg. In addition, the release of more relevant and reliable information could improve the image of the mining company, and it would lead to more confidence in the local fishes being safe for human consumption, especially among the people living in its vicinity. If the mining company can demonstrate that to be the case, it could also increase the social acceptance of mining activity in this area. In addition, many wild waterbirds and wild animals are adept at catching fish (e.g., perch), i.e., perch is an important part of the natural food chain. In principle, the environmental norm value for Hg has been set at 0.25 mg/kg for perch to protect fish-eating birds and some mammals [32]. Obviously, the lower the concentration of Hg and organic mercury in the environment, the better the wildlife is protected.

Mining products may be important in the future despite water contaminations by Hg and other toxic compounds. New methods for detecting methyl Hg [33,34,35] are welcome even though they are still in the developmental phase.

## 5. Conclusions

The weighted arithmetic mean (weighted according to fish weight) is a better way of inform consumers of fish about its safety. This type of calculation is more informative than other ways to calculate averages. In addition, it is the larger fish that can be the decisive factor in determining the actual Hg exposure in humans. Therefore, EFSA and other international institutes should recommend that, in the future, national health inspectors should use weighted means when examining the concentrations of pollutants in fish. Local health officers must protect humans from being exposed to high levels of Hg and in this way, they can promote the activity of local fishermen to sell their catch both locally and even internationally as it will be safe for human consumption.

Health officers should be familiar with many statistical tests. They need to be taught when, why and how to calculate the weighted means when estimating human exposure to Hg in fish. Therefore, they should receive detailed guidance on how this calculation should be carried out. The best way is to give examples like those listed in Table 1 and Table 2. Local health and environmental officers should be able to follow the scientific developments so that they know the reports of the EU, FAO and WHO [15,27]. In the future, environmental impact reports, such as [21], should include all individual Hg results, as well as the linked weights of the fish, such as was provided in [21], so that health and environmental research officials can also count the weighted means.

## Figures and Tables

**Table 1 ijerph-21-00758-t001:** The weights and lengths of the perch and their total mercury (Hg) concentration of the edible part (fillets) in two lakes [21].

Fish Weights (g)	Lengths (cm)	Total Hg Concentration in Fillet (mg/kg)
Lake Laakajärvi		
41	16.7	0.28
41	16.2	0.20
42	16.0	0.18
63	18.3	0.23
74	19.0	0.26
288	28.7	0.74
333	31.0	0.85
392	30.0	0.60
436	32.4	0.89
546	34.0	0.80
Lake Kiltua		
103	31.5	0.34
122	22.3	0.39
144	22.8	0.38
144	24.5	0.52
156	24.1	0.53
173	25.4	0.67
175	25.4	0.73
190	22.7	0.72
256	24.0	0.57
265	24.9	0.73

**Table 2 ijerph-21-00758-t002:** Different means of the Hg concentrations in the ten perches studied. The fishes were divided by the weights of the fish to estimate those exceeding or not exceeding the Hg value of 0.5 mg/kg. Their numbers and total amount of Hg in all ten perch samples, as well as the correlation coefficients between the amounts of Hg and the weights of the fish and correspondingly between the amounts of Hg and fish lengths in both Lakes Laakajärvi and Kiltua, are shown.

	Lake Laakajärvi	Lake Kiltua
Arithmetic mean ± standard deviation (mg/kg)	0.50 ± 0.30	0.55 ± 0.15
Geometric mean (mg/kg)	0.42	0.53
**Weighted mean (weighted according to fish weight) (mg/kg)**	**0.72**	**0.59**
Total weight of all ten perch (kg)	2.254	1.728
Total weight of perch containing less than 0.5 mg/kg Hg (+their number) (kg)	0.261 (5)	0.369 (3)
Total weight of perch containing more than 0.5 mg/kg Hg (+their number) (kg)	1.995 (5)	1.359 (7)
**Total amount of Hg in all ten fish (mg)**	**1.62**	**1.01**
Correlation coefficients between weight and Hg concentration	0.92	0.70
Correlation coefficients between length and Hg concentration	0.91	0.92

## Data Availability

The data presented in this study are available on request from the corresponding author.

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
