# Peer review of "Arithmetic vs. Weighted Means in Fish Fillets Mercury Analyses"

_ijerph, 2024, doi:10.3390/ijerph21060758_

Round 1

Reviewer 1 Report

Comments and Suggestions for Authors

Overall – this article is not publishable in its current form. The most important issue is that on first read there is a lack of an obvious aim to the paper. What problem is being addressed? What research advance is it establishing? Is it that weighted means should be used for fish consumption advisories? If so, this does not become evident until the midpoint of the narrative and does not even appear in the abstract. Also, while the format follows the established outline for a scientific paper: Background, Hypothesis/Aim, Methods, Result, Discussion, Conclusion, each section contains information that belongs in another section (see below). Also, the Discussion section needs to have a section on the weaknesses of the methods and uncertainty of the findings.

Title: the title of the paper should reflect the research aim or finding, such as: Variation in MeHg levels in Within Fish Species Analyses, or Arithmetic vs Weighed Means in Fish Tissue Hg Analyses, or something like that, if that is indeed the point of the paper (which as I stated, is not exactly clear).

1.      Introduction

Overall the paper is written at a very low level such as for a 13- or 14year old which is not the standard for scientific publications. Also, instead of each paragraph covering one topic, paragraphs often include varied bits of information that are not linked in a logical manner, including opinion statements which should not appear in the body of the paper until the Discussion section.

This section is too long and does not need to repeat the history of methyl mercury human exposure; hundreds of papers have been written on this topic.

The last paragraph before section 1.1 seems to summarize the problem addressed by this work and therefore should appear last in the Intro/Background section.

1.1  What is the point of including the information on sulphate/Hg metabolism at lake bottom? This information is not used to address Hg fish tissue analyses. This section includes statements about global mercury sources and deposition; why is that information in this paragraph and not in the first paragraph of the introduction?

1.2  The end of the first paragraph in this section seems to state the entire finding of this paper: Hg concentration in fishes can vary extensively – but mainly according to the size and age of the fish.  If this is already known then what is the point of this paper? The goal of the intro section is to show the reader you are addressing a gap in the literature. Also, this statement is not generally accepted as true. Fish consumption advisories give guidance according to species, not size and weight.

 The second paragraph in this section states an opinion and therefore should not appear in this paper until the Discussion. Or maybe the first sentence uses the word “should” incorrectly. 

The third paragraph in this section mentions a “Commission” but does not state what this commission is or does. 

The last paragraph in this section describes a new method for Hg analysis and then opines on whether this method is better. Why is this information included here? If it is not the method used in the analysis of fish in this paper it should either be removed or included in the Discussion section since it contains an opinion. And the opinion should have scientific data presented or referenced to substantiate the claim.

 2.      Materials and methods

First paragraph: a statement is made that if a waterway is not near an industrial source of Hg then the fish should be safe. This is not true. Due to global deposition all waterways are contaminated with mercury and many fish advisories pertain to waterways not located near industry.

Fourth paragraph: what is the point of the statement about size of fish making it difficult to obtain pure fillets? Is this a shortcoming in the method? If so, it should appear in a paragraph in the Discussion section about the weaknesses of the study. 

3.      Results

The Results section should be separate from the Discussion section. 

Table 1. What is the meaning of the word “Total” in the phrase “Total Hg concentration in fillet”? I assume this is simply the concentration measured in the individual fillet. If multiple levels were obtained from each fillet it should be made known in the Methods section. 

Table 2 label is a run-on sentence.

 The method for calculating weighted mean should be included in the Methods section.

 Second paragraph includes a statement that the company was happy about the results. This is an opinion and does not belong in a scientific paper unless substantiated by data. The fact that the company paid for the report belongs in the Discussion section not the Results section as it could be considered a weakness of this work.  Also, the information that the local population was not informed that some fish might have high Hg belongs in the Discussion Section not the Results.

 Third paragraph states it is impossible to estimate exposure from arithmetic or geometric means yet this is the method used to create fish advisories. Therefore, it is an opinion or a conclusion from the findings of this paper and therefore belongs in the Discussion/Conclusion section.

 Section 4.1

This section is not necessary to meet the aim of this research and may be better presented as an accompanying Letter to the Editor.

Comments on the Quality of English Language

Needs to be edited. There are multiple incidents of incorrect use of prepositions, making interpretation of the actual meaning difficult.

Author Response

Overall – this article is not publishable in its current form. The most important issue is that on first read there is a lack of an obvious aim to the paper. What problem is being addressed? What research advance is it establishing? Is it that weighted means should be used for fish consumption advisories? If so, this does not become evident until the midpoint of the narrative and does not even appear in the abstract. Also, while the format follows the established outline for a scientific paper: Background, Hypothesis/Aim, Methods, Result, Discussion, Conclusion, each section contains information that belongs in another section (see below). Also, the Discussion section needs to have a section on the weaknesses of the methods and uncertainty of the findings.

Thank you for your comments. The aim is now at the end of Introduction. The paper has been highly reduced.  

Title: the title of the paper should reflect the research aim or finding, such as: Variation in MeHg levels in Within Fish Species Analyses, or Arithmetic vs Weighed Means in Fish Tissue Hg Analyses, or something like that, if that is indeed the point of the paper (which as I stated, is not exactly clear).

Thank you! The new title is “Arithmetic vs Weighed Means in Fish Fillet Mercury analyses”, which is better.    

  1. Introduction

Overall the paper is written at a very low level such as for a 13- or 14year old which is not the standard for scientific publications. Also, instead of each paragraph covering one topic, paragraphs often include varied bits of information that are not linked in a logical manner, including opinion statements which should not appear in the body of the paper until the Discussion section.

This section is too long and does not need to repeat the history of methyl mercury human exposure; hundreds of papers have been written on this topic.

Thank you!  Introduction is now reduced. The history is now in 10 lines leading to the present limits of WHO & FAO and EU. 

The last paragraph before section 1.1 seems to summarize the problem addressed by this work and therefore should appear last in the Intro/Background section.

  • What is the point of including the information on sulphate/Hg metabolism at lake bottom? This information is not used to address Hg fish tissue analyses. This section includes statements about global mercury sources and deposition; why is that information in this paragraph and not in the first paragraph of the introduction?

Thank you! Methyl Hg is formed in lake (and sea) bottoms.  It is now as paragraph 1.1. Sulphate Reduction and Methyl Mercury formation.  

1.2  The end of the first paragraph in this section seems to state the entire finding of this paper: Hg concentration in fishes can vary extensively – but mainly according to the size and age of the fish.  If this is already known then what is the point of this paper? The goal of the intro section is to show the reader you are addressing a gap in the literature. Also, this statement is not generally accepted as true. Fish consumption advisories give guidance according to species, not size and weight.

 The second paragraph in this section states an opinion and therefore should not appear in this paper until the Discussion. Or maybe the first sentence uses the word “should” incorrectly. 

The third paragraph in this section mentions a “Commission” but does not state what this commission is or does. 

The last paragraph in this section describes a new method for Hg analysis and then opines on whether this method is better. Why is this information included here? If it is not the method used in the analysis of fish in this paper it should either be removed or included in the Discussion section since it contains an opinion. And the opinion should have scientific data presented or referenced to substantiate the claim.

Thank you! The weighed mean describes better the Hg content in fishes and thus it warns better the fish-eaters. The opinions are now in Conclusions.

  1. Materials and methods

First paragraph: a statement is made that if a waterway is not near an industrial source of Hg then the fish should be safe. This is not true. Due to global deposition all waterways are   contaminated with mercury and many fish advisories pertain to waterways not located near industry.

Fourth paragraph: what is the point of the statement about size of fish making it difficult to obtain pure fillets? Is this a shortcoming in the method? If so, it should appear in a paragraph in the Discussion section about the weaknesses of the study. 

Thank you! If the weight of perch would be less than 100 g, the weight of fillets is some 40 g and one fillet is 20 g. It is a difficult work to get this small bit of fish in an ordinary kitchen. Also for the companies the price of work for this 40 g may be too expensive. 

  1. Results

The Results section should be separate from the Discussion section. 

Table 1. What is the meaning of the word “Total” in the phrase “Total Hg concentration in fillet”? I assume this is simply the concentration measured in the individual fillet. If multiple levels were obtained from each fillet it should be made known in the Methods section.

Thank you! It is now described! I would understand the “Total Hg” as different inorganic and organic Hg-compounds including possible methyl-Hg compounds without specified the molecular structure. Methyl-Hg is an organic form with methyl group(s). 

Table 2 label is a run-on sentence.

 The method for calculating weighted mean should be included in the Methods section.

Thank you! Has been corrected.  

 Second paragraph includes a statement that the company was happy about the results. This is an opinion and does not belong in a scientific paper unless substantiated by data. The fact that the company paid for the report belongs in the Discussion section not the Results section as it could be considered a weakness of this work.  Also, the information that the local population was not informed that some fish might have high Hg belongs in the Discussion Section not the Results.

Thank you, has been omitted.

 Third paragraph states it is impossible to estimate exposure from arithmetic or geometric means yet this is the method used to create fish advisories. Therefore, it is an opinion or a conclusion from the findings of this paper and therefore belongs in the Discussion/Conclusion section.

Thank you! Has  moved and reduced and there is equation.

 Section 4.1

This section is not necessary to meet the aim of this research and may be better presented as an accompanying Letter to the Editor.

Comments on the Quality of English Language

Needs to be edited. There are multiple incidents of incorrect use of prepositions, making interpretation of the actual meaning difficult.

Thank you! OK but some preposition and other errors have been corrected. 

Reviewer 2 Report

Comments and Suggestions for Authors

This paper fills the knowledge gap when it comes to food safety with respect to fish consumption. 

Abstract: 

The authors did not state an aim or hypothesis of the paper in this section. 

Introduction

The introduction seems too long. Authors should be concise and only present relevant information that pertains to the aim of the paper. Authors should state a hypothesis for the paper in this section. 

Methodology: It should be concise and straight to the point. For example - Line 194 - 214 doesnt describe any methods, that section should be summarised if the aim is to give a background of the study sites. Methods were not adequately described. Example statistical analysis used were not descried and the methods for mercury determination was also not mentions. 

Discussion: The authors should work on making the discussion concise and straight to the point. 

Comments on the Quality of English Language

Good

Author Response

This paper fills the knowledge gap when it comes to food safety with respect to fish consumption. 

Thank you! I agree that there is a gap in knowledges.

Abstract: 

The authors did not state an aim or hypothesis of the paper in this section. 

Has been added. Thank you! 

Introduction

The introduction seems too long. Authors should be concise and only present relevant information that pertains to the aim of the paper. Authors should state a hypothesis for the paper in this section. 

 Thank you! It is now shorter, and the hypothesis with the aim is now at the end.

Methodology: It should be concise and straight to the point. For example - Line 194 - 214 doesnt describe any methods, that section should be summarised if the aim is to give a background of the study sites. Methods were not adequately described. Example statistical analysis used were not descried and the methods for mercury determination was also not mentions.

Thank you! The lines 194-214 have been moved. The means are now described.  

 Discussion: The authors should work on making the discussion concise and straight to the point. 

Thank you! It is now shorter and more concise.  

Comments on the Quality of English Languagecise

Good

Round 2

Reviewer 1 Report

Comments and Suggestions for Authors

Abstract still does not include aim of the paper and now does not match the title.

There is still no paragraph on weaknesses and uncertainty of the findings.

Comments on the Quality of English Language

Needs more editing.

Author Response

Thank you! The aim is now in two last sentences of abstarct. 

Thank you! The uncernity is now as the last paragraph of part 4.1. 

In addition, the reference 15 has got the journal.  There are small corrections also in lines 207 and 482.